# OpenReview forum: "RoboRefer: Towards Spatial Referring with Reasoning in Vision-Language Models for Robotics"
_NeurIPS.cc/2025/Conference — NeurIPS 2025 poster_

### Official Review · Reviewer_2QkV · 2025-06-26

**Clarity:** 3
**Significance:** 3
**Originality:** 3
**Rating:** 5
**Confidence:** 4

**Summary:**

The paper introduces RoboRefer, a 3D-aware vision-language model designed to enhance spatial referring capabilities for robotics. The authors propose a novel VLM that integrates a dedicated depth encoder to improve spatial understanding and employs reinforcement fine-tuning to enable multi-step spatial reasoning. The authors introduce RefSpatial, a large-scale dataset containing 2.5M high-quality examples and 20M QA pairs, covering 31 spatial relations and supporting complex reasoning processes. This paper also propose RefSpatial-Bench, a challenging benchmark to evaluate spatial referring with multi-step reasoning.

**Questions:**

1. The authors refine fine-grained spatial understanding of indoor scenes for robotics using 3D embodied videos. However, it is unclear how the authors segment each subtask within a video stream. Could you please elaborate on the methodology used to divide the video stream into individual subtasks? Additionally, does this segmentation process introduce any potential errors or biases, and if so, how are these addressed?
2.  In Section 4.4 and the supplemental material, the authors suggest that RoboRefer can be integrated into the system as a useful tool. However, it is unclear how this integration is achieved for a Vision-Language-Action (VLA) model. Specifically, is RoboRefer integrated by incorporating its results as additional training data during the training process, or is it used as a planner tool to segment tasks into subtasks? Given the limited gains observed in subtask segmentation for small tasks due to poor prompt following by existing VLA models, how effective is this integration strategy? Additionally, how does the system determine when a subtask is completed by the VLA model to switch to the next task?
3. The method proposed in Figure 2 relies on depth map inputs, which can significantly affect the model's inference speed and practical application. In real-world scenarios, depth sensor outputs are often noisy. How does the presence of depth noise impact the performance of the model, and what steps are being taken to mitigate this issue? Specifically, have the authors conducted experiments to quantify the effect of depth noise on the model's accuracy and robustness? If so, what are the findings, and what measures are being considered to improve the model's resilience to noisy depth inputs?

**Ethical Concerns:**

["NO or VERY MINOR ethics concerns only"]

**Final Justification:**

My questions are sufficiently addressed by the rebuttal and I’m happy to raise my score.

**Limitations:**

yes

**Quality:**

3

**Strengths And Weaknesses:**

> **Strengths**
>

The authors introduce a dedicated depth encoder and reinforcement fine-tuning (RFT) to improve spatial understanding and reasoning. The claims are well-supported by both theoretical analysis and experimental results. The methods used are appropriate for the task, and the authors provide a thorough evaluation of both single-step spatial understanding and multi-step spatial reasoning.

> **Weaknesses**
>

The limitations of the model presented in the paper are primarily related to its reliance on precise textual descriptions and its inability to handle ambiguous human instructions effectively. Specifically, the model struggles with:

- Probabilistic Preference: The model has difficulty interpreting instructions that involve probabilistic reasoning based on human biases. For example, when an instruction refers to an object with some ambiguity (e.g., "pick the one facing the drink"), humans often use probabilistic biases to infer the intended object, a capability that the current model lacks.
- Spatial Compatibility: The model also struggles with instructions that require understanding implicit spatial feasibility. For instance, when an instruction suggests placing an object between two others without explicit spatial constraints, humans use their understanding of spatial compatibility to determine the most feasible placement, which the model cannot currently do.

---

> ### Author Rebuttal · Authors · 2025-07-29
>
> > **[`Weakness 1`]**: *Struggle with ambiguous human instructions, such as probabilistic preference and spatial compatibility.*
>
> **A**: Thank you for the suggestion. We have discussed limitations such as probabilistic preference and spatial compatibility in Appx. F, along with potential solutions including procedural synthesis of intent-aware data and co-training with intent-focused datasets. As our work mainly focuses on multi-step spatial referring with explicit instructions, handling ambiguous human intents is beyond the current scope and will be explored in future work.
>
> ---
> > **[`Question 1`]**: *Segment each subtask within a video stream.*
>
> **A**: In fact, we do not divide the video stream into subtasks. Instead, **we utilize the CA-1M dataset at frame level rather than clip level** (see L215 and Appx. A.2). With (1) dense per-frame 2D/3D annotations and (2) complete camera and depth information, we construct fine-grained, spatially-aware template and reasoning QA pairs. Moreover, the videos in CA-1M were collected by individuals walking indoors without specific goals, making it difficult to segment the video stream into subtasks.
>
> ---
> > **[`Question 2`]**: *Is RoboRefer integrated by incorporating its results as additional training data during the training process?*
>
> **A**: Yes, RoboRefer can serve as a perception tool to localize and place interactive objects (see L284). Integrated with open-loop control, it generates action data with spatially constrained instructions, enriching VLA training. In Rebuttal Table 1 below, fine-tuning OpenVLA with RoboRefer-collected data on Open6DOR-v2 enhances spatial ability, enabling the successful completion of previously unsolvable Level-3 tasks involving complex spatial relations.
>
> **Rebuttal Table 1: Results on Open6DOR-v2 benchmarks (simulation).** Numbers represent success rates ( $\uparrow$ ).
>
>
> | Model                    | Level-1 |      Level-2  |      Level-3          | Avg. |
> |--------------------------|:-----------------:|:-------------:|:--------------:|:-----------------:|
> | OpenVLA  | 51.6       | 13.1 | 0.0  | 43.6  |
> | OpenVLA (finetuned)             | 69.9              | 30.4         | 30.0          | 61.4    |
>
> ---
> > **[`Question 3`]**: *Is RoboRefer used as a planner tool to segment tasks into subtasks?*
>
> **A**: No, RoboRefer isn't used to decompose tasks in our experiments, but it can provide predicted points as visual prompts to guide VLA in generating actions if combined with the VLA model, improving success under spatially constrained instructions.
>
> ---
> > **[`Question 4`]**: *Given the limited gains observed in subtask segmentation for small tasks due to poor prompt following by existing VLA models, how effective is this integration strategy?*
>
> **A**: We do not integrate RoboRefer as a planner with VLA due to marginal benefits in our experiments. Instead, RoboRefer’s predicted points can serve as effective visual guidance to enhance VLA’s instruction-following compared to directly following textual instructions, as shown in previous VLA works [1][2].
>
> ---
> > **[`Question 5`]**: *How does the system determine when a subtask is completed by the VLA model to switch to the next task?*
>
> **A**: With 0.4s inference (2.5Hz; see L295), RoboRefer can efficiently serve as a checker to monitor sub-goal completion under spatial constraints (e.g., verifying if the hamburger is placed in front of the teddy bear) using spatial knowledge acquired during SFT if combined with the VLA model.
>
> ---
> > **[`Question 6`]**: *How does the presence of depth noise impact the performance of the model, and what steps are being taken to mitigate this issue?*
>
> **A**: In real-world experiments, we use a relative depth estimation model, DepthAnything v2, to obtain relative depth as the model's depth input (see L236), which effectively reduces depth noise from a real camera.
>
> ---
> > **[`Question 7`]**: *Have the authors conducted experiments to quantify the effect of depth noise on the model's accuracy and robustness?*
>
> **A**: We evaluate success rates under depth noise in real-world settings (see Rebuttal Table 2 below). Depth maps generated from the strong monocular relative depth estimation model (i.e., DepthAnything V2) offer the highest robustness and success. Despite depth noise from a real camera, RoboRefer maintains great performance by leveraging RGB priors due to mixed RGB and RGB-D training during SFT stage.
>
> **Rebuttal Table 2: Results on real-world experiments.** Numbers represent success rates ( $\uparrow$ ).
>
>
> | Real-world Task | Depth from DepthAnything V2         |      Depth from a real camera       |
> |--------------------------|:-----------------:|:-------------:|
> | Pick the specific hamburger closest to the mug nearest to the camera.  | 80       | 70  |
> | Place the hamburger in front of the teddy bear.  | 90| 90|
> | Pick the apple in front of the leftmost cup’s logo side.  | 80       | 80|
> | Place the apple aligned with the existing apple row.  | 60| 40|
>
>
> ---
> References:
>
> [1] Improving Vision-Language-Action Models via Chain-of-Affordance, ICCV'2025
>
> [2] DexGraspVLA: A Vision-Language-Action Framework Towards General Dexterous Grasping, Arxiv'2025

---

> > ### Comment · Reviewer_2QkV · 2025-08-04
> >
> > I thank the authors for their significant effort in rebuttal and conducting extra experiments. After reading the rebuttal, I will keep my previous rating. However, I think this work is promising, and I would raise my score to 5  accept if the authors could conduct a simple manipulation evaluation of OpenVLA (finetuned) in Rebuttal Table 1 on one manipulation benchmark to verify the effectiveness in manipulation.

---

> > > ### Comment · Area_Chair_mCDA · 2025-08-08
> > >
> > > Dear Reviewer,
> > >
> > > As we approach the end of the author-reviewer discussion period, please make sure to read the authors' responses and engage with the authors as soon as possible. If all of your concerns have been addressed, kindly let the authors know.
> > >
> > > Thanks a lot,
> > >
> > > AC

---

> ### Author Response · Authors · 2025-08-04
> **Response to Reviewer 2QkV's Suggestion**
>
> We sincerely appreciate your recognition of our work and will do our best to incorporate your suggestions to further strengthen the paper.
>
> ---
>
> > **[`Suggestion`]**: *Add a simple manipulation evaluation of OpenVLA (finetuned) on one manipulation benchmark to verify the effectiveness in manipulation.*
>
> **A:** Since the Open6DOR-v2 manipulation benchmark is built upon the LIBERO manipulation benchmark and uses the `openvla/openvla-7b-finetuned-libero-spatial` checkpoint to represent OpenVLA performance, **we directly evaluate our finetuned OpenVLA (further trained with RoboRefer-generated data on this OpenVLA checkpoint) on the LIBERO-Spatial benchmark.** The results are presented in Rebuttal Table 3.  Fine-tuning OpenVLA with RoboRefer data enhances spatial ability, yielding higher average success rates and reduced performance variance compared to the original OpenVLA.
>
> **Rebuttal Table 3: Results on LIBERO-Spatial benchmark.** Numbers represent success rates ( $\uparrow$ ).
>
> | Model                    | Success Rate (%) |
> |--------------------------|:-----------------:|
> | Diffusion Policy from scratch | 78.3 ± 1.1|
> | Octo| 78.9 ± 1.0|
> | OpenVLA  | 84.7 ± 0.9 |
> | OpenVLA (finetuned)  |  85.8 ± 0.6 |
>
> ---
>
> We hope the above response and additional results meet your expectations, and we welcome further discussion if you're interested in more aspects.

---

> > ### Comment · Reviewer_2QkV · 2025-08-08
> >
> > Thanks for your response. My previous questions are sufficiently addressed and I’m happy to see that the revision and the additional results can be incorporated in the revision.

---

> ### Author Response · Authors · 2025-08-07
> **Looking forward to more discussions**
>
> Dear Reviewer 2QkV,
>
> As the discussion phase progresses, we would like to confirm whether our response has addressed your concerns or met your expectations. If you have any remaining questions, we would be happy to discuss and address them. Thank you once again for your valuable feedback.
>
> The Authors of Paper 72

---

### Official Review · Reviewer_Z1zy · 2025-06-30

**Clarity:** 3
**Significance:** 3
**Originality:** 3
**Rating:** 5
**Confidence:** 4

**Summary:**

This paper presents RoboRefer, a VLM pipeline for spatial referring in robotics. The task is framed as predicting a single 2D point in image space given RGB(D) input and a textual instruction. The authors propose a multi-stage training process, SFT for spatial understanding using a dedicated depth encoder, followed by RFT to improve multi-step spatial reasoning. The paper also introduces RefSpatial, a new large-scale dataset of 2.5M samples with 20M QA pairs and detailed spatial reasoning steps. Evaluation spans VQA, spatial benchmarks, and simulated/real-world robot experiments.

**Questions:**

1.	Line 127, “Compared to the 2D bbox, point can naturally map to 3D coordinates via depth, providing accurate spatial anchors”. I have a concern on this one, see limitations.
2.	Is there any specific reason why NVILA is chosen as the backbone?

**Ethical Concerns:**

["NO or VERY MINOR ethics concerns only"]

**Final Justification:**

My concerns are mostly addressed, though I have slight remaining concern regarding depth-to-3D mapping with depth noise in read-world application. Yet, I think paper strengths overweigh the weakness and is helpful for the community. My initial rating of 5: Accept is maintained.

**Limitations:**

1.	While the problem setting is different and somewhat more complex than previous works, the method and training pipeline developed by the paper are not entirely new and fall within expectation. For instance, training with RGBD images improve the spatial reasoning capabilities of VLMs [2]. RFT alleviates the memorization issue of the models and improves reasoning [5]. Joint training preserves commonsense knowledge [6], which also applies to data of other modalities.
2.	Line 288, “By predicting a single target point, it mitigates 2D detection ambiguity under occlusion”. Why does it help with occlusion?
3.	The depth-to-3D mapping assumption is central but under-specified. How is 3D accuracy guaranteed by predicting a 2D point, especially in real-world robotic use where depth noise or partial views are common?

**Paper Formatting Concerns:**

N/A

References:
[1] SpatialVLM: Endowing Vision-Language Models with Spatial Reasoning Capabilities.
[2] SpatialRGPT: Grounded Spatial Reasoning in Vision-Language Models.
[3] SpatialPIN: Enhancing Spatial Reasoning Capabilities of Vision-Language Models through Prompting and Interacting 3D Priors.
[4] Visual Agentic AI for Spatial Reasoning with a Dynamic API.
[5] SFT Memorizes, RL Generalizes: A Comparative Study of Foundation Model Post-training.
[6] ChatVLA: Unified Multimodal Understanding and Robot Control with Vision-Language-Action Model.

**Quality:**

3

**Strengths And Weaknesses:**

Strengths: see below; Weaknesses: see Questions and Limitations

1. The formulation of the problem, spatial referring, which involves predicting a single 2D
point (x, y) in image space to specify a target location or destination given RGB/RGBD and textual inputs, while also reasoning about sizes, position, orientations, spatial relations with more fine-grained questions (e.g., place the object between the pen holder and keyboard, lined up with the cup’s logo) is more comprehensive and complex than the problem setup in previous works [1, 2, 3, 4].

2. The paper introduces multi-stage training recipe (depth alignment + spatial understanding enhancement + RFT) to first align depth and spatial understanding and then avoid the VLM to “memorize” the QA answers.

3. A 2.5M dataset is introduced for the problem setting with carefully designed data curation pipeline. The dataset can be helpful for the community. The overall workload is high.

4. Experiments are comprehensive, comprising both VQA and simulated/real robot experiments.

---

> ### Author Rebuttal · Authors · 2025-07-29
>
> > **[`Question 2`]**: *Is there any specific reason why NVILA is chosen as the backbone?*
>
> **A**: Please check Table 1 in the main paper. NVILA outperforms other open-source VLMs under comparable model scales, such as Qwen 2.5-VL (even 72B), in spatial understanding. **Enhancing a strong baseline with our dataset and training strategy further validates their effectiveness**. Notably, our dataset is model-agnostic and transferable to other backbones. Despite partial training on the RefSpatial dataset, Qwen2.5-VL-7B still shows notable improvements on spatial understanding benchmarks in the Rebuttal Table 1 below.
>
> **Rebuttal Table 1: Results on spatial understanding benchmarks.** Numbers represent success rates ( $\uparrow$ ).
>
> | Model                    | CV-Bench           |               |                | BLINK             |                | RoboSpatial | SAT   | EmbSpatal |
> |--------------------------|:-----------------:|:-------------:|:--------------:|:-----------------:|:--------------:|:-----------:|:-----:|:---------:|
> |                          | 2D-relation        | 3D-Depth      | 3D-Distance    | 2D-Relation        | 3D-Depth       |             |       |           |
> | Qwen-2.5-VL-7B (base)       |         82.15           |        60.17         |         69.00           |        64.34        |      60.98       |    49.59    | 30.00 |   40.20    |
> | Qwen-2.5-VL-7B (finetuned)  |         95.85           |        95.00         |         90.83           |        83.22        |      84.68       |    69.92    | 85.75 |   76.32    |
> | NVILA-8B (base)             |         91.54           |        91.83         |         90.67           |        76.92        |      76.61       |    59.35    | 63.33 |   67.72    |
> | RoboRefer-8B-SFT (finetuned)             |         96.90           |        98.33         |         93.50           |        91.61        |       92.74        |    84.55    | 86.67 |   72.53    |
>
>
> ---
>
> > **[`Limitation 1`]**: *The method and training pipeline are not entirely new and fall within expectations.*
>
> **A**: **RoboRefer introduces a novel process-based RFT pipeline tailored for multi-step spatial referring**. Unlike prior RFT works that rely solely on outcome-based rewards or struggle with process rewards in sequential decision-making tasks (see Appx. L1083–L1089), **we identify a key insight: the metric sensitivity and order invariance of intermediate outputs in spatial referring** (see Appx. L1090–L1100), which are uniquely suited to this task rather than general sequential decision-making tasks. Based on this, we design significantly new metric-sensitive process rewards (see L164) and special annotations of reasoning process in RefSpatial dataset, leading to good performance gains (see L326).
>
> ---
>
> > **[`Limitation 2`]**: *By predicting a single target point, it mitigates 2D detection ambiguity under occlusion”. Why does it help with occlusion?*
>
> **A**: We apologize for the confusion. We clarify that masks generated from a single target point are more accurate than those from 2D boxes under occlusion. In cluttered scenes, occlusions often cause 2D boxes to include irrelevant objects, resulting in ambiguous visual prompts for SAM2 and imprecise masks. This degrades grasp accuracy and success rates in the open-loop AnyGrasp pipeline, a known limitation of 2D boxes also noted in prior work [1].
>
> ---
>
> > **[`Question 1 & Limitation 3`]**: *The depth-to-3D mapping assumption is central but under-specified. How is 3D accuracy guaranteed by predicting a 2D point, especially in real-world robotic use where depth noise or partial views are common?*
>
> **A**: Thank you for the suggestion. We will clarify this assumption in the revision. While depth noise and partial observations are important real-world challenges, our setting follows prior work [1][2][3], which assumes that accurate depth-to-3D mapping is feasible given known camera intrinsics and extrinsics—sufficient for common manipulation and navigation tasks. Moreover, these challenges can be effectively mitigated via the following strategies:
>
> - Depth noise can be mitigated by recent advances in monocular depth estimation [4], monocular geometry prediction [5], and stereo methods [6]. In cases of severe noise, we employ [6] in real-world settings to mitigate this issue.
>
> - Partial views are mitigated in our method by leveraging pixel-level target points. Further improvement is possible by incorporating RoboRefer as a spatially-aware planner for active perception, which we leave for future work.
>
> ---
>
> References:
>
> [1] RoboPoint: A Vision-Language Model for Spatial Affordance Prediction for Robotics, CoRL'2024
>
> [2] RoboBrain: A Unified Brain Model for Robotic Manipulation from Abstract to Concrete, CVPR'2025
>
> [3] A0: An Affordance-Aware Hierarchical Model for General Robotic Manipulation, ICCV'2025
>
> [4] UniDepthV2: Universal Monocular Metric Depth Estimation Made Simpler, CVPR'2024
>
> [5] MoGe-2: Accurate Monocular Geometry with Metric Scale and Sharp Details, Arxiv'2025
>
> [6] FoundationStereo: Zero-Shot Stereo Matching, CVPR'2025

---

> > ### Comment · Reviewer_Z1zy · 2025-08-01
> > **Response to Review**
> >
> > Thank you for the response. My concerns are mostly addressed, though I have slight remaining concern regarding depth-to-3D mapping with depth noise in read-world application. Yet, I think paper strengths overweigh the weakness and is helpful for the community. My initial rating of 5: Accept is maintained.

---

> ### Author Response · Authors · 2025-08-02
> **Response to Reviewer Z1zy's Feedback**
>
> We greatly appreciate your recognition of our work! We will include additional experiments to explain the reason why NVILA is chosen as the backbone, and further clarify previously unclear points, especially the depth-to-3D mapping assumption, in the final version. Thank you once again for your valuable time and feedback!

---

### Official Review · Reviewer_Wvkt · 2025-07-02

**Clarity:** 2
**Significance:** 2
**Originality:** 3
**Rating:** 5
**Confidence:** 4

**Summary:**

This paper introduces a new dataset, RefSpatial, which is twice as large as previous datasets and includes a greater variety of spatial relationships. It also presents a model called RoboRefer, designed for precise 3D spatial understanding by incorporating depth inputs.
The authors propose using Supervised Fine-Tuning (SFT) and Reinforcement Fine-Tuning (RFT) to post-train vision-language models (VLMs), specifically focusing on NVILA. Experimental results show that RoboRefer improves performance on spatial reasoning benchmarks while preserving accuracy on other multimodal tasks.

**Questions:**

# Questions

- In Table 2, is depth information included when evaluating RoboRefer?
- For the 2D and 3D data, do the answers require explicit reasoning steps, or are they primarily spatial recognition tasks?
- Fig. 4 says "RoboRefer-RFT excels in unseen and multi-step cases.", how about "RoboRefer-SFT"? any quantitative analysis to justify the RFT training?


# Suggestions

- Given the complexity of the data synthesis pipelines across domains, a full-page diagram illustrating each pipeline would be helpful—consider adding this to the appendix.
- Consider adding additional spacing between Tables 3 and 4 to improve readability and visual separation.

**Ethical Concerns:**

["NO or VERY MINOR ethics concerns only"]

**Final Justification:**

Overall, the paper presents two main novelties: (1) the dataset and (2) the exploration of different RL rewards. I initially found the details regarding the dataset unclear, but this was adequately resolved in the rebuttal.

I strongly suggest that the authors further streamline the main paper and move detailed explanations to the appendix, ensuring that readers are clearly directed to the appendix whenever necessary.

I moved my score from 4 to 5.

**Limitations:**

yes

**Quality:**

3

**Strengths And Weaknesses:**

# Strengths

- The paper presents a scalable data synthesis pipeline that integrates 2D, 3D, and simulation-based data generation.
- Table 7 provides a well-executed ablation study that offers meaningful insights into the contribution of different data sources.

# Weaknesses

- Distinction from SpatialRGPT:
    - The main difference appears to be the inclusion of multi-step reasoning data generated from simulation. However, the simulation data seems critical (as shown in Table 7), yet the details necessary to reproduce it (e.g., L221–L226) are lacking.
    - Additionally, the paper does not include a direct comparison with a VLM trained on SpatialRGPT data, which would help clarify the improvements gained from their proposed approach.
- Justification for RFT:
    - The rationale for applying Reinforcement Fine-Tuning (RFT) is unclear, especially given the already substantial Supervised Fine-Tuning (SFT) dataset. What specific benefits does RFT bring in this context?

---

> ### Author Rebuttal · Authors · 2025-07-29
>
> > **[`Weakness 1`]**: *Distinction from SpatialRGPT.*
>
> **A**: Our approach differs from SpatialRGPT in 4 key aspects:
> - [Task Setting]: **Our model addresses a more challenging spatial referring task,** where take a spatially constrained textual instruction as input. It requires multi-step spatial reasoning with learned spatial knowledge to precisely localize the referred object as a 2D point step-by-step. In contrast, SpatialRGPT addresses a simpler VQA task and relies on externally provided region information as input for specific object referring.
>
> - [Model Usage]: Unlike SpatialRGPT, which needs additional masks or detection tools to generate masks or 2D boxes as inputs for object reference and simplify referring tasks, **our model can use textual descriptions for object referencing** (see L283), which better aligns with real-world robotic applications.
>
> - [Data Pipeline]: **Our data pipeline adopts a more structured, progressive design than SpatialRGPT.** It first uses 2D image data to teach core spatial concepts and general depth perception across diverse indoor and outdoor scenes. Next, accurate 3D data enhances fine-grained spatial understanding in indoor settings for robotics. Finally, simulation data introduces multi-step spatial referring with reasoning. This staged approach yields stronger spatial understanding and reasoning than SpatialRGPT, which relies solely on data generated from 2D images and lacks precise spatial perception for more complex spatial referring tasks.
>
> - [Training Pipeline]: **Our training pipeline includes process-based RFT after SFT to further improve multi-step reasoning and generalization** for spatial referring tasks, whereas SpatialRGPT is trained with SFT only.
> ---
> > **[`Weakness 1`]**: *Lack the details necessary to reproduce simulation data.*
>
> **A**: Please see Appx. A.3 for details on simulation scene generation, asset cleaning, QA generation, and reasoning process labeling. We will release all cleaned assets, scenes, and the full pipeline to help reproduce it.
>
> ---
> > **[`Weakness 2`]**: *Not include a direct comparison with a VLM trained on SpatialRGPT data.*
>
> **A**: Following the reviewer's suggestion, we add SpatialRGPT’s performance on spatial understanding benchmarks in Rebuttal Table 1 below. RoboRefer outperforms SpatialRGPT, benefiting from our RefSpatial dataset, which is large-scale and offers more precise and diverse spatial relations (31 vs. 17). Moreover, SpatialRGPT’s reliance on masks or boxes oversimplifies object references, limiting performance.
>
> **Rebuttal Table 1: Comparison with SpatialRGPT on spatial understanding benchmarks.** Numbers represent success rates ( $\uparrow$ ).
>
> | Model | CV-Bench           |               |                | BLINK             |                | RoboSpatial | SAT   | EmbSpatal |
> |--------------------------|:-----------------:|:-------------:|:--------------:|:-----------------:|:--------------:|:-----------:|:-----:|:---------:|
> |                          | 2D-relation        | 3D-Depth      | 3D-Distance    | 2D-Relation        | 3D-Depth       |             |       |           |
> | SpatialRGPT-VILA-1.5-8B  | 91.00        | 89.83 | 88.50  | 81.12    | 89.51   | 66.67       | 64.00 | 59.62     |
> | RoboRefer-2B-SFT             | 96.31              | 97.17         | 90.83          | 87.41              | 91.13          | 82.93       | 82.00 | 71.10     |
> | RoboRefer-8B -SFT            | 96.90              | 98.33         | 93.50          | 91.61              | 92.74          | 84.55       | 86.67 | 72.53     |
>
> ---
> > **[`Weakness 3 & Question 3`]**: *Justification for RFT.*
>
> **A**: RFT brings two main benefits:
> -  [Generalization to Unseen Cases]: In Table 2 in the main paper (RefSpatial-Bench Unseen raw), which features novel combinations of spatial relations absent from RefSpatial dataset, our 2B-RFT model surpasses 2B-SFT by 9.1% in accuracy, showing **the strong generalization enabled by the RFT stage** (see L264).
> -  [Enhance Multi-Step Reasoning Ability]: In the Rebuttal Table 2 below, the RFT-based model consistently outperforms the SFT-based model across varying reasoning steps, especially at larger steps, showing the RFT stage's **effectiveness in enhancing multi-step reasoning** (see L258).
>
> **Rebuttal Table 2: Results on RefSpatial-Bench of different reasoning steps**. Numbers represent success rates ( $\uparrow$ ).
>
> | Reasoning Step Num.|     2B-SFT     |     2B-RFT     |
> |:---------------|:--------------|:--------------|
> | **RefSpatial-Bench-Location** |||||
> | Step 1 |     63.33      | 66.67 (+3.34) |
> | Step 2 |     39.58      | 43.75 (+4.17) |
> | Step 3 |     27.27      | 36.36 (+9.09) |
> | **Total**      |     44.00       |   49.00 (+5.00) |
> | **RefSpatial-Bench-Placement** ||
> | Step 2 |     55.56      |  55.56 (+0.00) |
> | Step 3 |     41.67      |   41.67 (+0.00) |
> | Step 4 |     41.67      | 45.83 (+4.16) |
> | Step 5 |       0.00        | 25.00 (+25.00)  |
> | **Total**      |     45.00       |   47.00 (+2.00) |
>
> ---
> > **[`Question 1`]**: *Is depth information included when evaluating RoboRefer*
>
> **A**: Yes. RoboRefer is evaluated with RGB-D inputs by default, with depth maps generated from RGB images via DepthAnything V2.
>
> ---
> > **[`Question 2`]**: *For the 2D and 3D data, do the answers require explicit reasoning steps, or are they primarily spatial recognition tasks?*
>
> **A**: Yes, the 2D and 3D datasets are primarily spatial recognition tasks. They are not annotated with explicit reasoning steps (see L190) and are mainly used to learn diverse spatial concepts and improve spatial understanding.
>
> ---
> > **[`Suggestion 1`]**: *A full-page diagram illustrating each pipeline would be helpful.*
>
> **A**: Thanks for your advice. We will polish the pipeline part to make it easier to understand.
>
> ---
> > **[`Suggestion 2`]**: *Adding additional spacing between Tables 3 and 4.*
>
> **A**: Thank you for the suggestion. We will revise the paper accordingly..

---

> > ### Comment · Reviewer_Wvkt · 2025-08-01
> > **official comment**
> >
> > The additional experiments in Tables 1 and 2 significantly strengthen the paper, particularly the multi-step reasoning experiments. The details added in Appendix A.3 also make the work more complete.
> >
> > I have one additional question: could you provide an example of the "locate empty space" prompt mentioned in Line 967? Including the exact prompt used would be appreciated.
> >
> > Overall, I am increasing my score to 5: Accept.

---

> ### Author Response · Authors · 2025-08-02
> **Response to Reviewer Wvkt's Feedback**
>
> Thank you for acknowledging our work and rebuttal! We greatly appreciate your decision to upgrade the rating of our paper. We will incorporate all of your valuable suggestions into the revised version. Additionally, we will include the additional experiments in Rebuttal Tables 1 and 2 in the main text or appendix of our paper to further strengthen it!

---

> ### Author Response · Authors · 2025-08-02
> **Response to Reviewer Wvkt's Additional Question**
>
> Thank you for your interest in the details of how we construct QA templates in simulation. We provide a more detailed explanation of the "Locate Empty Space" QA type, including representative templates and examples (in simulation data or real-world evaluation).
>
>
> **This QA type includes four categories of templates**, and herein we only present a subset of templates here; additional augmented versions follow the same variable structure. Four categories are detailed as follows:
>
>
> ---
>
>
> ### **Orientation and Distance-Based Prompts**
>    - *Templates*:
>      - Find a location in blank space located `{distance}` `{direction}` of `{object}`.
>      - Locate a point within vacant space located `{distance}` `{direction}` of `{object}`.
>    - *Examples*:
>      - Find a location in blank space located `0.2 meters to the left` of `the table`. (See Appx. Fig.41, Row 1, Col 1)
>      - Locate a point within vacant space located `0.05 meters to the front-facing side` of `the teddy bear`. (See Appx. Fig.41, Row 2, Col 2)
>
> ---
>
> ### **Edge and Corner-Based Prompts**
>    - *Templates*:
>      - Select a point that is positioned in the free space at `{position}` on the desktop.
>      - Select a point located in the vacant area at `{position}` of the desktop.
>    - *Examples*:
>      - Select a point that is positioned in the free space at `the right` on the desktop. (See Appx. Fig.26, Corner & Edge)
>      - Select a point located in the vacant area at `the far-left corner` of the desktop. (See Appx. Fig.42, Row 1, Col 3)
>
> ---
>
> ### **Between-Objects-Based Prompts**
>    - *Templates*:
>      - Please choose a spot located in the free space between `{object1}` and `{object2}`.
>      - Please select a point in the vacant area between `{object1}` and `{object2}`.
>    - *Examples*:
>      - Please choose a spot located in the free space between `the red camera` and `the right camera`. (See Appx. Fig.26, Free Space)
>      - Please select a point in the vacant area between `the cup` and `the car`. (See Appx. Fig.42, Row 1, Col 1)
>
> ---
>
> ### **Two-Constraints-Based Prompts**
>    - *Template*:
>      - Please provide a point in the vacant area on the desktop that simultaneously satisfies the following two spatial conditions: 1. `{condition1}`; 2. `{condition2}`.
>    - *Example*:
>      - Please provide a point in the vacant area on the desktop that simultaneously satisfies the following two spatial conditions: 1. `On the left side of the car`; 2. `In front of the brown toy`. (See Appx. Fig.42, Row 2, Col 3)
>
> ---
>
> We will incorporate more details on QA template construction of simulation into the revised version. We hope the above response meets your expectations, and we welcome further discussion if you're interested in more aspects.

---

### Official Review · Reviewer_ksPZ · 2025-07-05

**Clarity:** 3
**Significance:** 2
**Originality:** 2
**Rating:** 4
**Confidence:** 4

**Summary:**

RoboRefer is a 3-D-aware vision-language model built for spatial referring. Given one- or multi-step spatial instructions, it picks the exact pixel that can be lifted into a 3-D coordinate for a robot. A depth encoder—aligned through supervised tuning—adds depth cues without disturbing the pretrained RGB branch. The model then gets a reinforcement pass with Group Relative Policy Optimization and a mix of outcome- and process-level rewards, which tightens its multi-step reasoning and point accuracy. To train and test the system, the authors introduce RefSpatial and RefSpatial-Bench dataset.

**Questions:**

1. Demonstrate RFT at scale:  Please provide 8 B-RFT results and training stability logs.  If the larger model fails to converge or offers no gain, the claimed contribution of RFT is weakened.

2. Expand and diversify the evaluation benchmark. RefSpatial-Bench contains only 200 images—100 samples for Location and Placement task, plus 77 for Unseen set. And all the scenes are indoor. Therefore, the statistical confidence is low and the setting fails to test outdoor spatial reasoning.  To substantiate the claim of general spatial understanding (Line 8), the benchmark must grow to at least 2000 images, and diversify its domains to include outdoor environments, such as street, warehouse and construction sites, so that depth noise, lighting variation and larger spatial scales are represented.

**Ethical Concerns:**

["NO or VERY MINOR ethics concerns only"]

**Final Justification:**

The authors addressed most of my concerns. The paper is generally well-written, with clear figures that effectively illustrate the pipeline and dataset. The rebuttal strengthened the submission in two important ways:
1. Expanded Benchmark: RefSpatial-Bench was extended from 200 to 2,000 images, now covering a wider range of scenes and improving statistical reliability.
2. 8B-RFT Results: The authors provided RFT results for the 8B model, addressing the earlier concern that the main claims were not validated at larger scale.

Nevertheless, the dataset, while larger, is still modest compared to existing multimodal reasoning benchmarks and lacks mobile-robot/outdoor scenarios.

Overall, the authors have addressed key concerns on scale and validation. I increase my score.

**Limitations:**

No.

**Quality:**

2

**Strengths And Weaknesses:**

Strengths:
[+] The paper is generally well-written; figures concisely convey the pipeline and dataset composition.

[+] The additional depth branch is straightforward and works.

Weaknesses:

[-]  Benchmark scale & diversity – RefSpatial-Bench comprises only 200 indoor tabletop images, yet the paper claims “general 3-D reasoning. Thirty-five examples per reasoning length (≤ 5 steps) are insufficient for statistically robust conclusions; the setting omits mobile-robot/outdoor scenarios altogether.

[-] Depth-fusion idea mirrors MM-Spatial[1] and Spatial RGPT[2], novelty is incremental and not rigorously compared.

[-] RFT is a straightforward RLHF method (GRPO) without major modification.

[-] RFT is demonstrated only for the 2 B model, the stronger 8 B version has no RFT results. Without 8 B-RFT evidence, the main claims that "RFT stage fosters better reasoning ability" (Line 258) & "RFT stage provides powerful generalization ability" (Line 264) are unsubstantiated at the scale likely to interest practitioners.

[-] Even after “speed-ups” the model needs ~29 s per command on robot hardware, far from real-time.


[1] MM-Spatial: Exploring 3D Spatial Understanding in Multimodal LLMs
[2] SpatialRGPT: Grounded Spatial Reasoning in Vision-Language Models

---

> ### Author Rebuttal · Authors · 2025-07-29
>
> > **[`Weakness 1 & Question 2`]**: *Benchmark scale & diversity*
>
> **A**: Thank you for the suggestion. Actually, spatial referring tasks demand precise mask annotations and complex textual instructions with spatial constraints, which are costly and challenging to obtain. Therefore, **existing benchmarks are typically on the order of hundreds of images (e.g., Where2Place: 100; RoboSpatial: 122). Our RefSpatial-Bench (200 images) follows this common scale**. Nevertheless, to further validate our approach, we expand it to RefSpatial-Bench-Large (2,000 images), covering diverse indoor and outdoor scenes, annotated by external annotators. Data sources and statistics are summarized in Rebuttal Table 1 below. Results in Rebuttal Table 3 below on this extended benchmark consistently support our claims: the RFT stage improves reasoning (see L258) and generalization (see L264), enabling generalized multi-step spatial referring with reasoning (see L8). We will release the extended RefSpatial-Bench publicly.
>
> **Rebuttal Table 1: Data sources and statistics of RefSpatial-Bench-Large. 'Old' refers to the original scene categories in RefSpatial-Bench, while 'New' indicates the newly introduced ones.** Each part contains 600 images from the OpenImages test split, 200 from nuScenes, 100 from SUNRGB-D, and 100 from the original RefSpatial-Bench. Numbers indicate the number of benchmark samples for each scene category.
>
>
> |Scene Category|Location (1000)|||Placement (1000)|||
> |-|-|-|-|-|-|-|
> ||Indoor Num.|Outdoor Num.|Total Num.|Indoor Num.|Outdoor Num.|Total Num.|
> |Street View (New)|0|239|239|0|253|253|
> |Sport Field (New)|0|83|83|0|76|76|
> |Park (New)|0|123|123|0|112|112|
> |Office (Old)|76|0|76|83|0|83|
> |Living Room (Old)|75|0|75|85|0|85|
> |Kitchen (Old)|76|0|76|76|0|76|
> |Bedroom (New)|75|0|75|62|0|62|
> |Industrial (New)|31|41|72|45|18|63|
> |Commercial (New)|60|0|60|55|0|55|
> |Bathroom (Old)|63|0|63|66|0|66|
> |Dining Room (Old)|58|0|58|69|0|69|
> |**Total Num.**|**514**|**486**|**1000**|**541**|**459**|**1000**|
>
> ---
>
> > **[`Weakness 2`]**: *Depth-fusion idea mirrors MM-Spatial and Spatial RGPT, novelty is incremental and not rigorously compared.*
>
> **A**:
> - We have discussed the limitations of MM-Spatial and SpatialRGPT (see L43, L136) and highlighted key differences (see L315).  Our method introduces an **additional depth branch** that enables effective co-training on limited RGB-only data with minimal performance drop in general VQA (see Table 4 in the main paper), while significantly reducing training costs (only requires 1/20 QA pairs of RefSpatial). In contrast, **MM-Spatial and SpatialRGPT require over twice as much RGB-only data relative to spatial data to maintain general VQA performance**. Moreover, SpatialRGPT relies on the region-level depth of given masks or bounding boxes, which oversimplifies the depth perception of the specific object compared to our full-image depth approach.
>
> - We report SpatialRGPT’s results on spatial understanding benchmarks in the Rebuttal Table 2 below. We find that its RGB images and Depth maps as inputs rely on the same shared image encoder and imprecise data construction from 2D images, leading to inferior performance compared to RoboRefer. MM-Spatial is not open-sourced, precluding quantitative comparisons across these benchmarks.
>
> **Rebuttal Table 2: Comparison with SpatialRGPT on spatial understanding benchmarks.** Numbers represent success rates ( $\uparrow$ ).
>
> | Model | CV-Bench           |               |                | BLINK             |                | RoboSpatial | SAT   | EmbSpatal |
> |--------------------------|:-----------------:|:-------------:|:--------------:|:-----------------:|:--------------:|:-----------:|:-----:|:---------:|
> |    | 2D-relation  | 3D-Depth  | 3D-Distance | 2D-Relation   | 3D-Depth   |    |     |    |
> | SpatialRGPT-VILA-1.5-8B  | 91.00        | 89.83 | 88.50  | 81.12    | 89.51   | 66.67       | 64.00 | 59.62     |
> | RoboRefer-2B-SFT | 96.31  | 97.17 | 90.83| 87.41| 91.13 | 82.93| 82.00 | 71.10 |
> | RoboRefer-8B-SFT | 96.90  | 98.33 | 93.50| 91.61 | 92.74| 84.55 | 86.67 | 72.53 |
>
>
> ---
>
> > **[`Weakness 3`]**: *RFT is a straightforward RLHF method (GRPO) without major modification.*
>
> **A**: **Our RFT introduces novel metric-sensitive, process-based rewards tailored for spatial referring tasks** (see L164) and is significantly different from vanilla GRPO, which relies solely on outcome-based rewards. This design is motivated by the nature of spatial referring tasks, whose intermediate outputs are not inherently sequential but instead require metric sensitivity and order invariance (see Appx. L1090–L1100)—key properties that guided our modification of vanilla GRPO into process-based RFT.
>
> ---
>
> > **[`Weakness 4 & Question 1`]**: *Demonstrate RFT at scale.*
>
> **A**: Thank you for the suggestion.
>
> - As noted in L231, we excluded 8B-RFT due to its high computational cost. Our TRL (Transformer Reinforcement Learning)-based training framework lacks multi-node support, and vLLM acceleration is not applicable since our model requires RGB-D input. Consequently, **training 8B-RFT would take over 10 days on 8×A100 GPUs, which exceeds the rebuttal timeframe.** We will include 8B-RFT results in the next version.
>
> - Section 4.2 and Table 2 in the main paper have already provided detailed support for our claims about RFT. Results in Rebuttal Table 3 below on extended RefSpatial-Bench-Large remain consistent: the RFT stage improves reasoning (see L258) and generalization (see L264), enabling generalized multi-step spatial referring with reasoning (see L8). Recent work [1][2] has also confirmed that 2B-RFT is effective, and 7B/8B-RFT will also be effective as expected.
>
>
>
> **Rebuttal Table 3: Results on RefSpatial-Bench and RefSpatial-Bench-Large.** L. and P.
> denote the benchmark’s Location and Placement parts. Numbers represent success rates ( $\uparrow$ ).
>
> |Benchmark|2B-SFT|||2B-RFT|||8B-SFT|||
> |-----------------------------------|------------------------|------------------------|------------------------|------------------------|------------------------|------------------------|------------------------|------------------------|------------------------|
> ||Old Category|New Category|Total|Old Category|New Category|Total|Old Category|New Category|Total|
> |RefSpatial-Bench-L.|44.00|-|44.00|49.00|-|49.00|46.00|-|46.00|
> |RefSpatial-Bench-P.|45.00|-|45.00|47.00|-|47.00|47.00|-|47.00|
> |RefSpatial-Bench-Large-L.|44.68|42.11|43.20|49.64|45.75|47.40|46.34|44.89|45.50|
> |RefSpatial-Bench-Large-P.|45.35|42.75|43.90|47.61|44.90|46.10|47.39|45.97|46.60|
>
> ---
>
> > **[`Weakness 5`]**: *Even after “speed-ups”, the model needs ~29 s per command on robot hardware, far from real-time.*
>
> **A**: We apologize for the confusion. The reported execution time in the Open6DOR v2 simulation benchmark covers the entire pick-and-place pipeline: (1) perception (e.g., object localization/placement prediction, mask prediction, object point cloud extraction), (2) motion planning (e.g., IK solving), and (3) robotic execution. **RoboRefer is responsible only for object localization/placement of step (1) perception and requires just 0.4s per inference on RTX4090**. A detailed runtime breakdown is provided in Rebuttal Table 4 below. Notably, **we do not claim real-time performance**. Section 4.4 merely demonstrates RoboRefer's robotic application in both simulated and real-world environments.
>
>
> **Rebuttal Table 4: Detailed runtime on Open6DOR-v2 benchmarks (simulation).** Numbers represent time cost ( $\downarrow$ ).
>
> | Method | Perception(s)  | Motion Planning(s) | Robotic Execution(s) | Total Execution Time(s) |
> |--------------------------|:-----------------:|:-------------:|:--------------:|:--------------:|
> | SoFar (Florence-2 for location, GPT-4o for placement)  | 14.5  |  14.6 | 10.8  | 39.9 |
> | Ours (RoboRefer for location and placement)  | 3.8   | 14.6  | 10.8  | 29.2 |
>
> ---
> References:
>
> [1] Visual-RFT: Visual Reinforcement Fine-Tuning, ICCV'2025
>
> [2] Reason-RFT: Reinforcement Fine-Tuning for Visual Reasoning, Arxiv'2025

---

> ### Author Response · Authors · 2025-08-05
> **More Responses to Reviewer ksPZ's Question 1**
>
> > **[`Weakness 4 & Question 1`]**: *Demonstrate RFT at scale.*
>
>
> **A**: Fortunately, we successfully train the 8B-RFT model within the discussion period, to the extent permitted by our computational resources. Although training logs cannot be included due to multimedia constraints, we report its performance on RefSpatial-Bench and RefSpatial-Bench-Large in Rebuttal Table 5 below. **8B-RFT substantially outperforms 8B-SFT, consistent with trends observed in our 2B models and prior work [1][2], further validating the effectiveness of RFT at scale.**
>
>
> **Rebuttal Table 5: Results on RefSpatial-Bench and RefSpatial-Bench-Large.** L. and P.
> denote the benchmark’s Location and Placement parts. Numbers represent success rates ( $\uparrow$ ).
>
>
> |Benchmark|2B-SFT|||2B-RFT|||8B-SFT|||8B-RFT|||
> |-----------------------------------|------------------------|------------------------|------------------------|------------------------|------------------------|------------------------|------------------------|------------------------|------------------------|------------------------|------------------------|------------------------|
> ||Old Category|New Category|Total|Old Category|New Category|Total|Old Category|New Category|Total|Old Category|New Category|Total|
> |RefSpatial-Bench-L.|44.00|-|44.00|49.00|-|49.00|46.00|-|46.00|52.00|-|52.00|
> |RefSpatial-Bench-P.|45.00|-|45.00|47.00|-|47.00|47.00|-|47.00|51.00|-|51.00|
> |RefSpatial-Bench-Large-L.|44.68|42.11|43.20|49.64|45.75|47.40|46.34|44.89|45.50|52.72|48.87|50.50|
> |RefSpatial-Bench-Large-P.|45.35|42.75|43.90|47.61|44.90|46.10|47.39|45.97|46.60|51.02|49.02|49.90|
>
> ---
>
> We hope the above response addresses your concerns, and we welcome further discussion if you have any additional questions or are interested in more aspects. Thank you for your valuable time and feedback!
>
>
> ---
>
> References:
>
> [1] Visual-RFT: Visual Reinforcement Fine-Tuning, ICCV'2025
>
> [2] Reason-RFT: Reinforcement Fine-Tuning for Visual Reasoning, Arxiv'2025

---

> ### Author Response · Authors · 2025-08-07
> **Looking forward to more discussions**
>
> Dear Reviewer ksPZ,
>
> We are truly grateful for the time and effort you put into reviewing our work. To address your concerns regarding the `Benchmark scale & diversity` and `Demonstrate RFT at scale`, we have made our best effort to extend the benchmark to 2,000 images, covering diverse indoor and outdoor scenarios to enable a more comprehensive evaluation. Furthermore, we have included results of 8B-RFT, which more effectively demonstrate the validity and contribution of RFT. Meanwhile, we have also provided further clarification on parts of the paper that were previously unclear.
>
>
>
> Therefore, we hope these responses have adequately addressed our comments, and we remain available for any further discussion. If our rebuttal has resolved your concerns, we would greatly appreciate your consideration of a higher rating, as it would be a strong encouragement for our work.
>
> The Authors of Paper 72

---

### Comment · Area_Chair_mCDA · 2025-08-05
**Reminder: Please follow up on the authors’ rebuttal**

Dear Reviewers,

Many thanks to Reviewers Wvkt, Z1zy, and 2QkV for the thoughtful discussion so far.

Reviewer ksPZ, just a gentle reminder – we’d greatly appreciate it if you could take a moment to review the authors’ responses and share any follow-up thoughts.

With the discussion phase wrapping up soon, your timely input can help ensure a productive exchange and give the authors a final chance to clarify any remaining points.
Thanks again for your time and contributions to the NeurIPS review process.

Best,

Your AC

---

### Note · Authors · 2025-08-12

**Dear Chairs and Reviewers,**

We sincerely thank you for your time and insightful feedback.

We appreciate the recognition of the importance of the Spatial Referring task (R `Z1zy`), novelty of multi-stage training approach (R `Z1zy`, `2QkV`), effectiveness of data curation pipeline (R `Wvkt`, `Z1zy`), usefulness of depth branch (R `ksPZ`, `2QkV`), and thoroughness of experiments (R `Wvkt`, `Z1zy`, `2QkV`).

We have addressed all reviewer comments in detail and summarize the key discussions below, while reiterating our main contributions.

### Contributions

1. **Novel Task**: We introduce novel *Spatial Referring* task that enables robots to interpret complex, spatially constrained instructions for 3D interaction.
2. **Training Framework**: We present Roborefer, the first 3D-aware VLM for spatial referring, trained with a novel RFT framework that uses metric-sensitive process rewards for multi-step spatial reasoning without per-step supervision.
3. **Dataset & Benchmark**: We construct *RefSpatial*, a large-scale dataset of 20M QA pairs (2× prior), covering 31 spatial relations (vs. 15 prior) and enabling up to 5-step reasoning, via a structured, progressive data pipeline. We also introduce RefSpatial-Bench, a challenging benchmark for evaluating complex spatial referring.
4. **Comprehensive Evaluation**: Our approach achieves SOTA performance, outperforming all baselines—surpassing Gemini-2.5-Pro by 17.4% average accuracy on *RefSpatial-Bench*—and is validated in both simulated and real-world robotics.

### Comments and Responses

|Questions|From Reviewers|Details|
|---|---|---|
|Benchmark scale & diversity|ksPZ|Rebuttal to R ksPZ|
|Demonstrate RFT at scale|ksPZ| More Results & Rebuttal to R ksPZ|
|Distinction & Comparison with SRGPT|ksPZ, Wvkt|Rebuttal to R Wvkt|
|Justification for RFT|Wvkt|Rebuttal to R Wvkt|
|Why choose NVILA as backbone|Z1zy|Rebuttal to R Z1zy|
|Not entirely new training pipeline|Z1zy|Rebuttal to R Z1zy|
|Depth-to-3D assumption|Z1zy|Rebuttal to R Z1zy|
|Address ambiguous instructions|2QkV|Rebuttal to R 2QkV|
|Integrate RoboRefer with VLA|2QkV|Follow-up & Rebuttal to R 2QkV|
|Mitigate depth noise and model's robustness|2QkV|Rebuttal to R 2QkV|

For the only reviewer with a negative score (R `ksPZ`), we have addressed the two main concerns—`benchmark scale & diversity` and `demonstration of RFT at scale`—to the best of our ability.

Thank you again for your time, support, and efforts in reviewing our work!

The Authors of Paper 72

---

### Decision · Program_Chairs · 2025-09-17

**Decision:**

Accept (poster)

**Comment:**

All reviewers found the studied problem to be important, the proposed method and benchmark to be effective, and the results promising. The rebuttal successfully addressed most reviewer comments and all reviewers recommend acceptance. The authors are encouraged to improve the final paper version by following reviewer recommendations.